# Role of Histone Variant H2A.J in Fine-Tuning Chromatin Organization for the Establishment of Ionizing Radiation-Induced Senescence

**DOI:** 10.3390/cells12060916

**Published:** 2023-03-16

**Authors:** Mutaz A. Abd Al-razaq, Benjamin M. Freyter, Anna Isermann, Gargi Tewary, Adèle Mangelinck, Carl Mann, Claudia E. Rübe

**Affiliations:** 1Department of Radiation Oncology, Saarland University Medical Center, 66421 Homburg/Saar, Germany; 2Institute for Integrative Biology of the Cell (I2BC), CEA, CNRS, Université Paris-Saclay, 91198 Gif-sur-Yvette, France

**Keywords:** histone variant H2A.J, ionizing radiation, radiation-induced senescence, Senescence-Associated Heterochromatin Foci (SAHF), DNA-Segments with Chromatin Alterations Reinforcing Senescence (DNA-SCARS), Senescence-Associated Secretory Phenotype (SASP)

## Abstract

Purpose: Radiation-induced senescence is characterized by profound changes in chromatin organization with the formation of *Senescence-Associated-Heterochromatin-Foci* (SAHF) and *DNA-Segments-with-Chromatin-Alterations-Reinforcing-Senescence* (DNA-SCARS). Importantly, senescent cells also secrete complex combinations of pro-inflammatory factors, referred as *Senescence-Associated-Secretory-Phenotype* (SASP). Here, we analyzed the epigenetic mechanism of histone variant H2A.J in establishing radiation-induced senescence. Experimental Design: Primary and genetically-modified lung fibroblasts with down- or up-regulated H2A.J expression were exposed to ionizing radiation and were analyzed for the formation of SAHF and DNA-SCARS by immunofluorescence microscopy. Dynamic changes in chromatin organization and accessibility, transcription factor recruitment, and transcriptome signatures were mapped by ATAC-seq and RNA-seq analysis. The secretion of SASP factors and potential bystander effects were analyzed by ELISA and RT-PCR. Lung tissue of mice exposed to different doses were analyzed by the digital image analysis of H2A.J-immunohistochemistry. Results: Differential incorporation of H2A.J has profound effects on higher-order chromatin organization and on establishing the epigenetic state of senescence. Integrative analyses of ATAC-seq and RNA-seq datasets indicate that H2A.J-associated changes in chromatin accessibility of regulatory regions decisively modulates transcription factor recruitment and inflammatory gene expression, resulting in an altered SASP secretome. In lung parenchyma, pneumocytes show dose-dependent H2A.J expression in response to radiation-induced DNA damage, therefore contributing to pro-inflammatory tissue reactions. Conclusions: The fine-tuned incorporation of H2A.J defines the epigenetic landscape for driving the senescence programme in response to radiation-induced DNA damage. Deregulated H2A.J deposition affects chromatin remodeling, transcription factor recruitment, and the pro-inflammatory secretome. Our findings provide new mechanistic insights into DNA-damage triggered epigenetic mechanisms governing the biological processes of radiation-induced injury.

## 1. Introduction

Senescence is a stress response that limits the replication of damaged or aged cells by implementing permanent cell cycle arrest [1]. Senescent cells display profound changes in nuclear and chromatin organization with the formation of Senescence-Associated Heterochromatin Foci (SAHF). Importantly, senescent cells also secrete complex combinations of predominantly pro-inflammatory factors, collectively referred to as Senescence-Associated Secretory Phenotype (SASP) [2]. Replicative senescence was first discovered in cultured fibroblasts, where prolonged passaging and replicative exhaustion led to growth arrest due to critically short telomeres [3]. Premature senescence can also be triggered in response to intrinsic and extrinsic stressors, such as acute DNA damage after exposure to ionizing radiation (IR). Double-strand breaks (DSBs) represent the most deleterious type of radiation-induced DNA lesions [4]. Subsequent DNA damage response (DDR) mechanisms coordinate cellular DSB repair activities by recruiting multiple repair proteins such as 53BP1 to break sites, forming radiation-induced DNA repair foci [5]. Severe or irreparable DNA damage can cause persistent 53BP1-foci, detectable for days and weeks after their formation, suggesting that DSB repair mechanisms are incapable of resolving these lesions [6]. In senescent cells, persistent DNA damage with accumulated DDR proteins, such as promyelocytic leukaemia (PML) nuclear bodies [7], form nuclear substructures called DNA-Segments with Chromatin Alterations Reinforcing Senescence (DNA-SCARS) [8,9]. Together with SAHF, DNA-SCARS are one of the most prevalent nuclear markers of cellular senescence.

Senescent cells undergo dramatic alterations to their chromatin landscape that affect genome accessibility and transcriptional programs [10]. Nucleosomes, the basic units of chromatin, are composed of 147 base pairs of DNA wrapped around the histone octamer core. The octamer contains four heterodimers of the canonical histone proteins H2A, H2B, H3, and H4. Linker histone H1 binds to internucleosomal DNA to stabilize higher-order chromatin structures and participate together with non-histone proteins in the dynamic regulation of chromatin compaction. Canonical histones are the most abundant in nucleosomes and are synthesized and incorporated in a replication-dependent manner. Additional diversity is provided by the incorporation of histone variants into chromatin [11]. Histone variants can profoundly change chromatin properties by modulating nucleosome stability and function, which may affect their interaction with chromatin remodelers and modifiers [12]. The deposition of canonical histones is coupled to DNA synthesis, whereby canonical histones assemble into nucleosomes behind replication forks. The incorporation of histone variants, in contrast, occurs throughout the cell-cycle and is independent of DNA synthesis. In non-cycling senescent cells, canonical histone production declines and variant histones tend to accumulate. The incorporation of histone variants can have direct effects on the structure and stability of nucleosomes, thereby modifying the accessibility of DNA and regulating the whole transcriptome by the dynamic binding of transcription factors (TF). Deciphering the underlying regulatory networks that balance the open/active and closed/repressed chromatin states and thereby modulating gene expression programmes may help to identify key regulators of radiation-induced senescence.

Previous studies have shown that histone variant H2A.J accumulates in human fibroblasts in vitro as well as in murine and human skin tissue in vivo during replicative, oncogene- and radiation-induced senescence, and affects the inflammatory gene expression of senescent cells [13,14,15]. In this study, we focused on the epigenetic mechanisms of H2A.J modifying transcription programmes during radiation-induced senescence progression and describe how these deregulations eventually contribute to radiosensitivity or radioresistance.

## 2. Materials and Methods

### 2.1. Cultured Lung Fibroblasts

Primary WI-38 human lung fibroblasts were obtained from ATCC. Immortalized WI-38 hTERT-fibroblasts were genetically modified to prepare H2A.J knock-down (H2A.J-KD) or knock-in (H2A.J-KI), canonical H2A knock-in (H2A-KI) and non-targeted controls (NT) as described previously [13]. Stable fibroblast populations were cultured at 5% O_2_ and 5% CO_2_ in MEM (Invitrogen, Karlsruhe, Germany) with 10% fetal bovine serum, 1 mM sodium pyruvate, 2 mM L-glutamine, 0.1 mM non-essential amino acids, and 1% penicillin/streptomycin. A total of 1 µg/mL doxycycline was added to medium for 1 week prior to IR to express shRNA sequences or to overexpress H2A or H2A.J. The cells were homogenously grown on coverslips and used for experiments once 90% confluency was achieved.

### 2.2. Radiation Exposure of Cultured Lung Fibroblasts

Cells were exposed to IR using the linear accelerator Artiste™ (Siemens, Munich, Germany) (6-MV photons; dose-rate 2 Gy/min). Cells were analyzed at different time-points (5 h, 24 h, 1 w and 2 w post-IR) following IR (20 Gy) and compared to non-irradiated controls (non-IR).

### 2.3. Immunofluorescence Microscopy (IFM) Analysis

Cells were fixed with 4% paraformaldehyde and permeabilised with 0.5% Triton X-100, washed with 0.1% Tween^®^-20, and incubated overnight with a primary antibody (anti-H2A.J, ActiveMotif, Waterloo, Belgium; anti-53BP1, Bethyl, Montgomery, TX, USA; anti-p21, Abcam, Berlin, Germany; anti-PML, Santa Cruz, CA, USA) followed by Alexa-Fluor^®^488 or Alexa-Fluor^®^568 secondary antibody (Invitrogen, Karlsruhe, Germany). Subsequently, cells were mounted in a VECTAshield™ mounting medium with 4′, 6-diamidino-2-phenylindole (DAPI, Vector Laboratories, Burlingame, CA, USA). Fluorescence images were captured with a Nikon-Eclipse Ni fluorescence-microscope equipped with a Nikon DS-Qi2 camera (Nikon, Düsseldorf, Germany). For evaluating H2A.J-, 53BP1-, SAHF-, and p21-positivity, at least 200 cells were captured for each sample (positive cells in %). For quantifying 53BP1-foci per cell, at least 50 foci and/or 50 cells were analyzed per sample. For the identification of DNA-SCARS, 53BP1/PML co-localization was analyzed in 50 nuclei, and numbers of 53BP1/PML co-localizing foci relative to total 53BP1-foci were expressed in percentages.

### 2.4. Immunhistochemistry (IHC) Analysis (SA-β-Gal)

Following 5 min fixation with 2% paraformaldehyde and 0.2% glutaraldehyde, cells were incubated with X-Gal staining solution (AppliChem GmbH, Darmstadt, Germany) at 37 °C overnight. After 30 s of methanol incubation, the dried samples were permeabilised with 0.2% Triton X-100 and washed with 1% BSA. Samples were blocked with 4% BSA for 1 h, followed by an overnight incubation with H2A.J primary antibody. Incubation with Dako immunoglobulin/bioatinylated secondary antibody (Agilent, Waldbronn, Germany) was followed by Vectastain ABC Peroxidase standard (Vector, Burlingame, CA, USA) and SIGMAFAST™ 3.3′Diaminobenzidine (Merck, Darmstadt, Germany) incubations, respectively. Samples were finally mounted in Dako Faramount Mounting Medium (Agilent, Waldbronn, Germany).

### 2.5. RNA-Sequencing (RNA-Seq)

RNA extraction, library preparation, and paired-end (2 × 150 bp, 30 M reads per sample) sequencing using NovaSeq (Illumina, San Diego, CA, USA) were performed by GENEWIZ Germany GmBH (Leipzig, Germany). Subsequently, raw reads were trimmed with Cutadapt (v3.5) to clip low-quality ends from reads that were below a Phred quality score of 20. Additionally, a minimum read length of 50 was set. Data quality was inspected before and after trimming using FastQC (v0.11.3) [16]. Sequences were aligned to the human genome sequence GRC38.p13 [17] using the Burrows-Wheeler Alignment Tool (v0.7.17) [18] and counted with HTSeq (v0.13.5) [19]. Subsequently, a differential expression analysis was performed using DESeq2 (v1.34.0) [20].

### 2.6. Assay for Transposase-Accessible Chromatin (ATAC-Seq)

WI-38 hTERT-fibroblasts were washed and incubated with 1xDNase (Worthington Biochemical Corp., Lakewood, NJ, USA). Tagmentation, library preparation, and paired-end (2 × 42 bp, 10 M reads per sample) sequencing using NovaSeq500 (Illumina, San Diego, CA, USA) were performed by ActiveMotif (Carlsbad, CA, USA). The obtained sequences were aligned to the human genome sequence GRC38.p13 [17] using BWA (v0.7.12) [18], and PCR duplicates were removed. Next, peaks were called using MACS2 (v2.1.0) [21] and normalized by random sampling and tag number reduction to the tag number of the smallest sample. Subsequently, overlapping intervals were grouped into merged regions, and differentially accessible sites were identified with DESeq2 (v1.34.0) [20]. Finally, motif analysis for the 2500 regions with the highest accessibility differences was performed using HOMER (v4.11) [22].

### 2.7. Integrative Analysis of ATAC-Seq and mRNA-Seq Data

Fragments per million mapped fragments were obtained with the fpm function of DESeq2 [20] and visualised as bar graphs using GraphPad Prism 9 (GraphPad Software, San Diego, CA, USA). Dot plots visualizing HOMER [22] differential chromatin accessibility analysis for each relevant condition were prepared using the ggplot2 (v3.3.5) R package. BAM files of appropriate samples were merged using Bedtools (v2.23.2) [23] and sorted with Samtools (v1.14) [24]. Subsequently, bedgraph files were obtained and normalized to RPKM using bamCoverage from deepTools (v3.4.3) [25] and converted to BigWig format with Bedgraphtobigwig (v357). Subsequently, tracks were visualized using Gviz (v1.38.3) with the support of biomaRt (v2.50.3) for gene locations, and tidyverse (v1.3.1) R packages, combining expression data, chromatin accessibility, differentially accessible regions obtained with MACS2 and DESeq2 (differential accessibility adjusted *p* value ≤ 0.1), and motif locations from HOMER analysis.

### 2.8. Enzyme-Linked ImmunoSorbent Analysis (ELISA)

Conditioned media (CM) were collected from three independent cultures of NT, H2A.J-KD and H2A.J-KI fibroblasts. CM from cultured fibroblasts was collected at 2 weeks after 20 Gy IR compared to non-irradiated fibroblasts. Cell numbers in flasks were determined at the timepoint of the CM collection. CM samples were frozen at −80 °C until analysis. A Multi-Analyte ELISArray™ Kit (Qiagen, Hilden, Germany) was used to screen SASP factors with a standard ELISA plate reader, according to the supplier’s protocol.

### 2.9. Reverse-Transcriptase Quantitative Polymerase-Chain-Reaction (RT-qPCR) analysis

RNA extraction was completed using TRIzol™ (ThermoFisher Scientific, Darmstadt, Germany) and phenol/chloroform phase separation. A QuantiTect^®^ Reverse Transcription Kit (Qiagen, Hilden, Germany) was used for cDNA synthesis and Advanced Universal SYBR Green Supermix (Bio-Rad Laboratories GmbH, Feldkirchen, Germany) was used for subsequent quantitative PCR in a CFX96 Touch Real-Time PCR Detection System (Bio-Rad Laboratories GmbH), as previously described [14].

### 2.10. Radiation Schedule and Lung Tissue Sampling in Mice

C57BL/6-N mice (2-month-old) were purchased from Charles River Laboratories (Sulzfeld, Germany) and housed in pathogen-free rooms to minimize infection risks, and supplied with a standard laboratory diet and water ad libitum. For whole-body irradiation (6-MV photons; dose-rate, 2 Gy/min) using an Artiste™ linear accelerator (Siemens, Munich, Germany), the animals were placed in an 18 cm-diameter Plexiglass cylinder covered by 1.5 cm thick plastic to improve dose homogeneity. Computed-tomography-based three-dimensional dose calculations were made with the Pinnacle planning system (Philips Radiation Oncology Systems, Fitchburg, WI, USA). Animals were irradiated with single-doses (10 Gy or 2 Gy) or with fractionated IR (once daily from Monday to Friday: 5 × 2 Gy, 20 × 0.1 Gy). At defined time-points after the last IR exposure, mice were anesthetized, perfused, and the lungs were removed, fixed overnight in 4% paraformaldehyde (Sigma-Aldrich Chemie GmbH, Munich, Germany), and processed for further analysis. At least three biological (experimental animals) and three technical (whole lung slices) replicates were examined for each irradiation regimen. The experimental protocol was approved by Medical Sciences Animal Care and Use Committee of Saarland University.

### 2.11. IFM and IHC Analysis of Lung Tissue Sections

Formalin-fixed tissues were embedded in paraffin and sectioned at 4-μm thickness. After deparaffinization in xylene and rehydration in decreasing alcohol concentrations, sections were boiled in a citrate buffer (Dako Agilent pathology solutions, Santa Clara, CA, USA) and incubated with Roti-Immunoblock (Carl Roth, Karlsruhe, Germany). For IFM analysis, sections were incubated with primary antibodies (anti-H2A.J, Active Motif, Carlsbad, CA, USA) followed by AlexaFluor-488 secondary antibody (Invitrogen, Karlsruhe, Germany) and mounted in VECTAshield with 4′,6-diamidino-2-phenylindole (DAPI; Vector Laboratories, Burlingame, CA, USA). For IHC analysis, sections were incubated with anti-H2A.J antibody followed by biotin-labeled antibodies (Dako, Glostrup, Denmark) and staining was completed by incubation with 3,3′-diaminobenzidine and substrate chromogen. Finally, sections were counterstained with haematoxylin and mounted with Aqueous-Mounting-Medium (Dako, Glostrup, Denmark). For qualitative analysis, H2A.J-positive cells were visualized under the Nikon E600 epifluorescent microscope (Nikon, Düsseldorf, Germany).

### 2.12. Digital Image Analysis of H2A.J-Immunohistochemistry

Whole digital slide images were obtained from H2A.J-stained lung sections of irradiated (5 × 2 Gy; 24 h post-IR) and non-irradiated mice using Axioscan 7 (ZEISS, Oberkochen, Germany). In lung parenchyma, 10 representative regions were annotated manually (excluding bronchiolar ducts and blood vessels) using the multiplex IHC module of HALO^®^ image analysis software (version 3.4.2986; IndicaLabs, Albuquerque, NM, USA). In these regions of interest (ROI), the brown DAB staining signal for H2A.J was compared to the hematoxylin counterstain for each nucleus, and cells were categorised into no (blue), weak (yellow), moderate (orange) and strong (red) H2A.J-stained nuclei. The percentage of cells was calculated for each staining category in relation to the total cell numbers in ROIs.

### 2.13. Statistical Analysis

GraphPad Prism (version 9.4.1, GraphPad Software, San Diego, CA, USA) was used to collect and analyse data. Data are presented as the mean of three experiments ±SEM. One-way analysis of variance (ANOVA) with Dunnett’s multiple comparisons test was used for comparison among different groups. A *p* value of <0.05 was considered statistically significant, <0.01 as statistically highly significant, and <0.001 as extremely statistically significant. Significant statistical differences compared to non-irradiated controls (marked by asterisks alone) or between cell lines (asterisks with square brackets) are presented in the figures as * (*p* < 0.05), ** (*p* < 0.01), and *** (*p* < 0.001).

## 3. Results

### 3.1. H2A.J Accumulation and Radiation-Induced Senescence 

Immortalized fibroblasts with H2A.J knock-down (H2A.J-KD) or knock-in (H2A.J-KI) and canonical H2A knock-in (H2A-KI) were analyzed compared to non-targeted controls (NT). Confluent H2A.J-KD, H2A.J-KI, H2A-KI and NT fibroblasts were irradiated with 20Gy, and H2A.J expression was analyzed at 24 h, 1 w and 2 w post-IR by IFM. For NT fibroblasts, we observed time-dependent increases of pan-nuclear H2A.J staining intensities (Figure 1A, left panel). The quantification of H2A.J+ cells revealed steady increases from ≈10% in non-irradiated (non-IR) to ≈80% in irradiated NT (2 w post-IR) (Figure 1A, right panel). While H2A.J-KD fibroblasts showed nearly no staining (≤6% H2A.J+ cells), H2A.J-KI fibroblasts revealed an intense pan-nuclear H2A.J staining for all analyzed time-points (90–100% H2A.J+ cells). Non-irradiated H2A-KI fibroblasts displayed slightly higher H2A.J expression levels (≈18%) than non-irradiated NT fibroblasts (≈10%); however, after IR exposure, the proportion of H2A.J+ cells increased to nearly 80%. Collectively, these findings indicate an effective and functioning cell system to study the pathophysiological role of H2A.J in radiation-induced senescence. The most widely used biomarker for aging cells is senescence-associated β-galactosidase (SA-β-Gal). Double-staining for H2A.J (nuclear brown signal) and SA-β-Gal (cytoplasmic blue signal) was established to analyze these senescence markers simultaneously (Figure 1B, left panel). SA-β-Gal activity measured in non-irradiated NT, H2A.J-KD, H2A.J-KI and H2A-KI fibroblasts showed generally low numbers of SA-β-Gal+ cells (≈10%). After IR exposure, NT, H2A.J-KD and H2A-KI fibroblasts revealed an increased expression of lysosomal β-galactosidase protein, and proportions of SA-β-Gal+ cells clearly increased to ≈95% at 2 w post-IR, suggesting that these fibroblast populations properly enter the senescence state (Figure 1B; right panel). In H2A.J-KI fibroblasts, by contrast, SA-β-Gal induction was observed in only ≈50% of fibroblasts at 1 w and 2 w post-IR, suggesting that high proportions of these cells failed to enter senescence (Figure 1B; right panel). Stringent cell growth arrest associated with cellular senescence can be determined by augmented levels of cyclin-dependent kinase inhibitor proteins, such as p21^Cip1/Waf1^. Accordingly, p21 staining was established to verify our finding of reduced senescence induction in H2A.J-KI fibroblasts after IR exposure. While all non-irradiated fibroblast populations revealed very low p21 levels (≤7%), the proportion of p21+ cells increased to 75–90% for NT, H2A.J-KD and H2A-KI after IR, reflecting their strong senescence response. Significantly, for H2A.J-KI fibroblasts, the p21 expression level remained very low even after IR exposure (≈14%), suggesting that H2A.J-KI fibroblasts are largely resistant to radiation-induced senescence.

### 3.2. Formation of SAHF and DNA-SCARS following IR

The dynamic reorganization of the higher-order chromatin structure during senescence progression leads to the formation of dense, repressive heterochromatin foci, so-called SAHFs, visible via DAPI staining. To investigate the input of H2A.J expression to global chromatin organization, we quantified the number of SAHF+ cells at 2 w post-IR compared to their non-irradiated counterparts (Figure 2A). In NT and H2A.J-KD fibroblasts, SAHFs emerged in ≈70% of the cells after IR exposure. For H2A-KI fibroblasts, the proportion of SAHF+ cells was significantly lower (≈40%), and in H2A.J-KI fibroblasts the number of SAHF+ cells was reduced to only ≈10%, despite serious genotoxic insults induced by IR (Figure 2A). Previous studies have shown that, in contrast to the even genome-wide distribution of canonical H2A, the histone variant H2A.J predominantly localizes in distinct chromatin regions with persistent DNA damage and activated DDR mediators [14]. To test whether this specific deposition pattern of H2A.J is correlated with the formation of DNA-SCARS, double-staining for 53BP1 and PML was performed in NT, H2A.J-KD, H2A.J-KI and H2A-KI fibroblasts, before and 5 h, 24 h and 2 w post-IR (Figure 2B). For all analyzed cell lines, very low 53BP1-foci levels were observed in non-irradiated fibroblasts (0.1–1.8 foci/cell), but clearly higher 53BP1-foci levels were observed after IR exposure, with 42–50 foci/cell at 5 h post-IR, 12–15 foci/cell at 24 h post-IR, and 5–7 foci/cell at 2 w post-IR (Figure 2B). This similar decline of 53BP1-foci in H2A.J-gene depleted and overexpressed fibroblasts suggests, that the DSB repair capacity is not affected by differential H2A.J expression. Enumerating PML-foci per cell, we observed low foci levels (6–12 foci/cell) in all non-irradiated controls, slightly higher levels at 5 h and 24 h post-IR (10–17 foci/cell), but clearly higher values at 2 w post-IR, particularly in the H2A.J-KI and H2A-KI lines of fibroblasts (40–46 foci/cell) (Figure 2B). PML nuclear bodies are involved in genome maintenance pathways including DDR and DNA repair and p53-associated apoptosis. H2A.J knock-down did not affect the formation of PML-foci; H2A.J knock-in, however, led to an increased formation of PML-foci after IR compared to their NT counterpart. Quantitative scoring of 53BP1/PML-double-positive foci revealed clearly lower co-localization events in H2A.J-KI cells, and thus reduced the numbers of DNA-SCARS (Figure 2B). Collectively, these findings suggest that differential H2A.J deposition in chromatin may have profound effects on higher-order chromatin organization and on establishing the epigenetic state of senescence in response to DNA-damaging IR.

### 3.3. ATAC-Seq for Chromatin Accessibility Analysis

Open and closed chromatin configurations regulate gene expression through regulatory networks of transcription factors (TF), which modulate transcription by recognizing and binding to specific DNA sequences. Changes in chromatin accessibility for TF binding sites were studied by ATAC-seq, permitting the genome-wide analysis of transcriptionally open/active and closed/repressed regulatory elements. Using the motif discovery tool HOMER, the sequence-based prediction for TF motif frequency or activity was mapped and compared for different conditions: for each line of fibroblasts in their post-IR versus non-IR status (Figure 3A) and between different fibroblast lines for their non-IR (Figure 3B) and post-IR (Figure 3C) status. ATAC-seq data analysis identified numerous members of the Activator Protein-1 (AP-1) superfamily (ATF3, BATF, FOSL2, JUNB) as the most affected motifs in differentially accessible chromatin regions of NT fibroblasts after IR exposure (Figure 3A). The dimeric AP-1 complex is composed of different members of ATF, FOS and JUN families, thereby providing multiplicity for regulatory control. Our findings correlate with previous studies in WI-38 lung fibroblasts, showing that these predominant transcription factors act as pioneers for imprinting the transcriptional programme of senescent cells [26]. H2A.J-KD and H2A.J-KI cells, in contrast, revealed less open chromatin configurations for these TF motifs after IR exposure (Figure 3A, left panel). Significantly, for H2A.J-KI fibroblasts, these chromatin regions are even less accessible after IR exposure, which may correlate with their increased resistance to radiation-induced senescence (Figure 3A, right panel). To address the functional importance of H2A.J in chromatin accessibility, these enriched TF motifs were profiled in NT, H2A.J-KD and H2A.J-KI fibroblasts, compared in each case for their non-IR and post-IR conditions. For these motifs, H2A.J-KI versus NT and, even more pronounced, H2A.J-KI versus H2A.J-KD fibroblasts revealed increasingly open chromatin in both, their non-IR (Figure 3B, left panel) and post-IR status (Figure 3C, left panel). Compared to this, H2A.J-KD fibroblasts clearly exhibit more closed chromatin configurations for these TF binding sites (Figure 3C, right panel). Collectively, our findings suggest that deregulated H2A.J expression compromised the dynamic response capacity for TF recruitment.

### 3.4. Integrative Analysis of ATAC-Seq and RNA-Seq Datasets

Transcriptionally active genes are characterized by accessible chromatin at their promoters, nearby enhancers, and some internal regions within the gene body itself. To determine whether differentially accessible chromatin configurations were correlated with their corresponding gene expression levels, ATAC-seq and mRNA-seq datasets were integrated for NT, H2A.J-KD and H2A.J-KI fibroblasts in their irradiated versus non-irradiated state. Figure 4 shows the integrated results from ATAC-seq and RNA-seq analysis for the previously identified signature genes interleukin-6 (IL6), colony-stimulating factor 2 (CSF2), CC-chemokine ligand 2 (CCL2), and C-X-C motif ligand 8 (CXCL8) (Figure 4). Strikingly, we observed clear differences between these genetically-modified fibroblast populations regarding their inflammatory gene regulation after IR exposure. By comparing ATAC-seq with RNA-seq data for NT fibroblasts, increased chromatin accessibility (green marks) after IR was correlated with increased IL6 mRNA signals. In contrast with H2A.J-KD fibroblasts, reduced chromatin accessibility (red marks) after IR was associated with the down-regulation of IL6 mRNA expression. In contrast with H2A.J-KI fibroblasts, different chromatin regions showed reduced or increased accessibility, ultimately leading to the overall down-regulation of IL6 mRNA expression. Similar correlations between ATAC-seq and RNA-seq data can be observed for the other immune-modulatory cytokines. Collectively, the integration of ATAC-seq and RNA-seq results indicate that changes in chromatin accessibility were clearly associated with inflammatory gene expression. 

RNA-seq analysis was performed in NT, H2A.J-KD and H2A.J-KI fibroblasts for measuring gene expression changes in response to IR exposure. Volcano plots show unchanged, up- and down-regulated genes in NT, H2A.J-KD, and H2A.J-KI fibroblasts at 2 w post-IR, compared to their non-irradiated controls (Appendix A). During the progression of radiation-induced senescence, the chromatin structure changes fundamentally, and the proportion of canonical histones and histone variants changes significantly (Appendix A). The precise distribution of histones and histone variants is crucial for chromatin organization and its epigenetic states.

### 3.5. Senescence-Associated Secretory Phenotype (SASP)

After high-dose IR exposure fibroblasts become senescent, but remain viable for long intervals and develop SASP with the secretion of inflammatory cytokines. For all lines of fibroblasts (most pronounced for NT fibroblasts) transcribed SASP genes (IL6, CSF2, CCL2, CXCL8) were generally upregulated following IR compared to their non-irradiated counterparts. Subsequently, transcribed SASP genes were correlated with corresponding SASP proteins by investigating the secretion of interleukin-6 (IL6), granulocyte-macrophage colony-stimulating factor (GM-CSF), monocyte chemoattractant protein-1 (MCP1), and interleukin-8 (IL8) in culture media by ELISA, as readout for the inflammatory secretome. Non-irradiated NT, H2A.J-KD and H2A.J-KI fibroblasts expressed only low levels of these common SASP factors, and hence no senescence-messaging secretome (Figure 5B). NT fibroblasts in radiation-induced senescence, by contrast, secreted high levels of IL6, GM-CSF, MCP1 and IL8, demonstrating that SASP components were significantly increased between non-senescent and senescent states in NT fibroblasts (Figure 5B). Strikingly, H2A.J-KD and also H2A.J-KI fibroblasts revealed distinctly lower protein expression levels for most of these SASP factors in radiation-induced senescence (Figure 5B). The only exception was IL8 in H2A.J-KI fibroblasts, with clearly increased expression levels in non-irradiated H2A.J-KI compared to NT fibroblasts, and with further significant increases after IR exposure. In summary, our findings indicate that the genetically-modified H2A.J expression in H2A.J-KD and H2A.J-KI fibroblasts differentially modulates the various soluble signaling factors of SASP with similar expression patterns to those in RNA-seq experiments. Next, we explored whether SASP components enriched in culture media (CM) of senescent cells may affect paracrine functions in non-irradiated cells (Figure 5C). Naive NT fibroblasts were treated with CM from non-senescent or senescent cells (CM from NT fibroblasts: non-IR versus post-IR) for periods of 24 h or 72 h, respectively, and the relative gene expression of these SASP factors (IL6, CSF2, CCL2, CXCL8) was monitored by RT-qPCR (Figure 5C). A gene expression analysis of NT fibroblasts revealed the significantly increased mRNA expression of IL6 and CXCL8 (with up to ~5-fold increases; Figure 5C) after treatment with CM from senescent cells. Our results show that treatment with senescent CM induced paracrine effects on naive cells and resulted in secondary SASP induction with significantly elevated transcription levels of proinflammatory cytokines. These findings suggest that senescent lung fibroblasts affect their local environment and induce so-called ‘bystander’ effects through the secretion of bioactive SASP components.

### 3.6. H2A.J Expression in Lung Tissue after IR Exposure

Previous studies noted organ-specific differences in H2A.J protein expression in normal tissues generally, with strong labeling of luminal epithelial cells, even without IR exposure [27]. Here, we used IFM to visualize H2A.J expression within different cell types in lung tissue of mice exposed to different doses of IR. Lung parenchyma are comprised of greatly varying cell populations with bronchiolar and alveolar epithelium (comprising type I and II pneumocytes), alveolar macrophages, and endothelial and interstitial cells. In non-irradiated lung tissue, the bronchial epithelium and some type II pneumocytes, characterized by more spherical shapes and known to release pulmonary surfactant to lower surface tension, stained positive for H2A.J. After IR with moderate doses (single-dose or fractionated IR with 2 Gy), we observed clear increases in the number of H2A.J-positive pneumocytes, most pronounced at 24 h post-IR (Figure 6A). After high-dose exposure with 10 Gy (24 h post-IR), we observed striking increases of H2A.J expression in the alveolar epithelium and in multiple other cell types, including interstitial fibroblasts, scattered throughout the lung parenchyma. Moreover, we observed cytosolic H2A.J staining after high-dose IR, which was most pronounced in the bronchiolar epithelium. The meaning of this finding is currently unclear, but recent studies suggest that histones are released into the extracellular space after significant cellular damage, thereby triggering thrombus formation and innate immunity during acute tissue injury [28]. Previous studies have shown that acute exposure to high-dose IR disrupts the alveolo-capillary barrier and increases cytokine release, resulting in pulmonary edema and promoting the recruitment of inflammatory cells [29,30]. As a proof-of-concept, we applied automated whole-slide imaging and high-resolution image analysis of chromogenic H2A.J-staining to show that clinically relevant fractionation schemes result in increased H2A.J expression in lung tissue. In representative regions of irradiated (5 × 2 Gy, 24 h post-IR) versus non-irradiated lung parenchyma, cells were classified into no (blue), weak (yellow), and moderate.

**Figure 4 cells-12-00916-f004:**
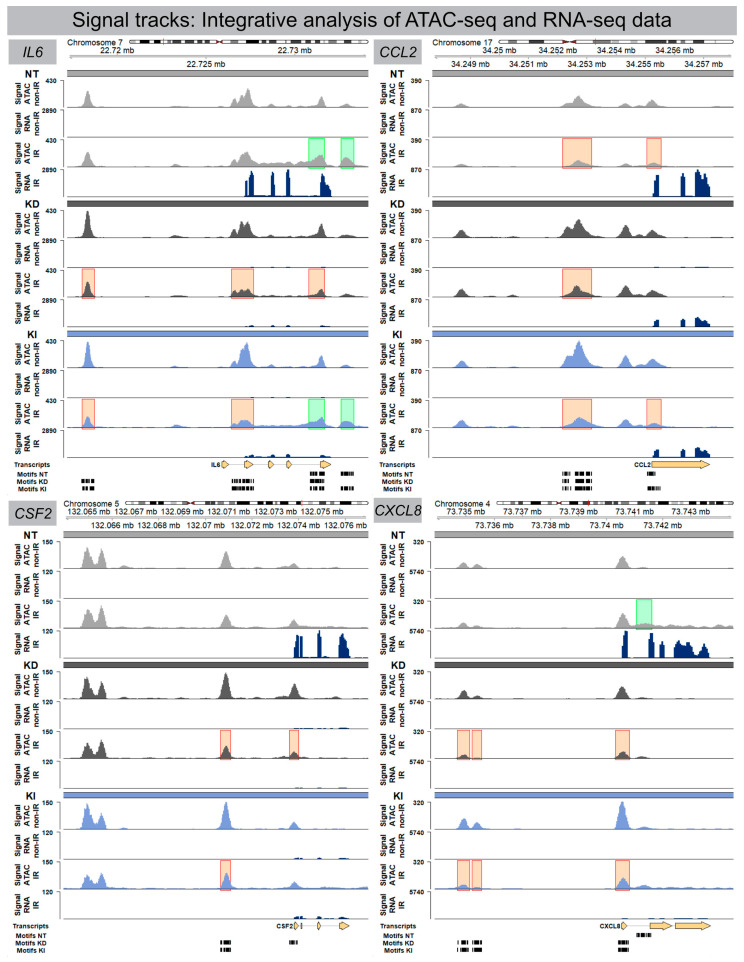
Signal track integration for SASP gene expression. Region-specific analysis of chromatin accessibility (ATAC-seq) and RNA expression (RNA-seq) for SASP factors IL6, CSF2, CCL2 and CXCL8 in NT, H2A.J-KD, and H2A.J-KI fibroblasts at 2 w post-IR, compared to non-irradiated controls. Colour-coded boxes covering ATAC-seq signal tracks in irradiated conditions indicate more (green) or less (red) accessible chromatin regions (compared to non-irradiated controls) and depicts a strong correlation with RNA-seq signals (blue), (orange), and *strong* (red) H2A.J-stained nuclei (Figure 6B). After automatic classification, the percentage of cells with different nuclear H2A.J-staining intensities was quantified in relation to total cell numbers. Our quantitative results show an overall increase of H2A.J+ cells from ≈20 to ≈40% after IR exposure; in addition, we observed an increase of H2A.J+ cells within each category, reflecting that H2A.J expression is significantly enhanced after IR exposure (Figure 6B). Collectively, our findings suggest that dose-dependent H2A.J expression triggered by the initial exposure to IR may contribute to the initiation and progression of radiation-induced lung injury.

**Figure 5 cells-12-00916-f005:**
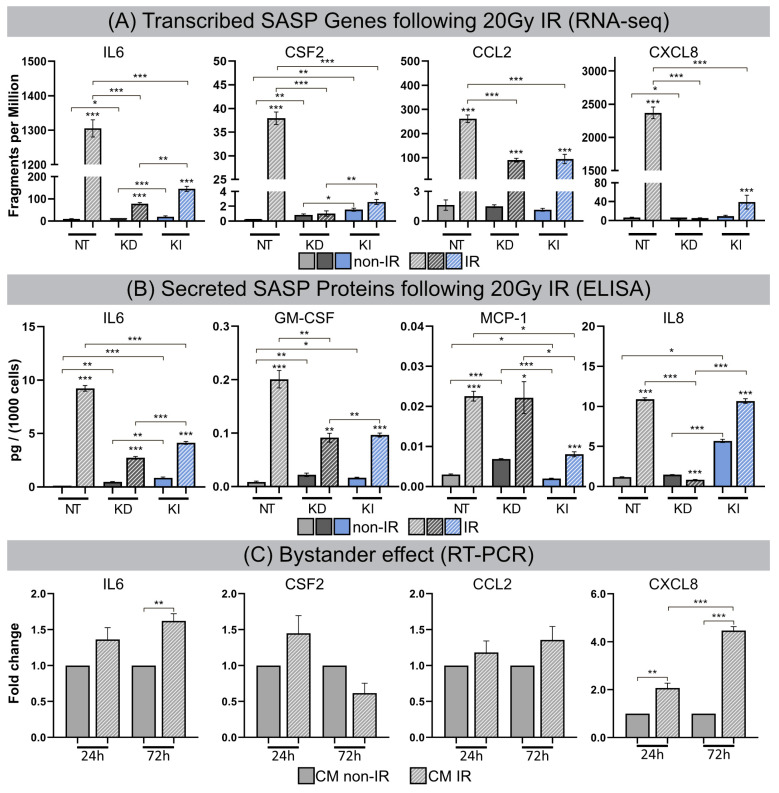
(**A**) Transcribed SASP genes following 20 Gy IR. Quantification of mRNA expression (mRNA-seq) for IL6, CSF2, CCL2 and CXCL8 in NT, H2A.J-KD, and H2A.J-KI fibroblasts at 2 w post-IR, compared to non-irradiated controls. (**B**) Secreted SASP proteins following 20 Gy IR. Quantification of protein expression (ELISA) for IL6, GM-CSF, MCP-1, IL8 in conditioned medium of NT, H2A.J-KD, and H2A.J-KI fibroblasts at 2 w post-IR, compared to non-irradiated controls. (**C**) Bystander effect. Quantification of mRNA expression (RT-PCR) for IL6, CSF2, CCL2 and CXCL8 in NT fibroblasts after 24 h- or 72 h-exposure to conditioned medium of irradiated (CM IR) or non-irradiated (CM non-IR) NT fibroblasts. Significant statistical difference compared to non-irradiated controls (marked by asterisks alone) or between cell lines (asterisks with square brackets): * *p* < 0.05; ** *p* < 0.01; *** *p* < 0.001.

**Figure 6 cells-12-00916-f006:**
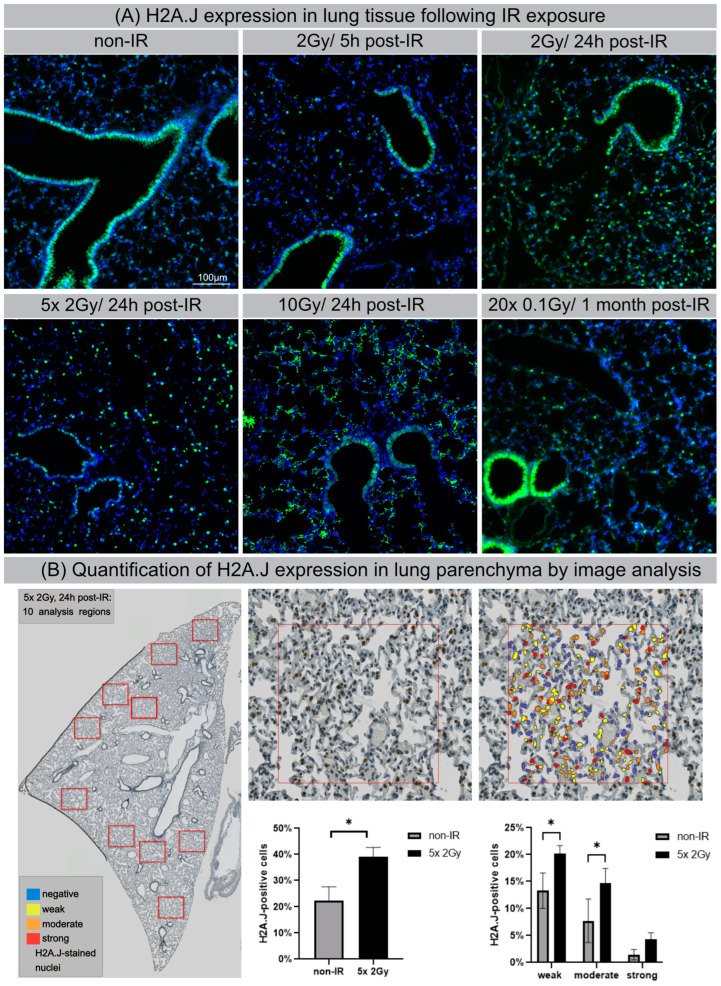
H2A.J expression in lung tissue after IR exposure (**A**) IFM micrographs of H2A.J expression in lung tissue after single-dose exposure with 2 Gy (5 h and 24 h post-IR) or 10 Gy (24 h post-IR) and after fractionated IR with 5 × 2 Gy or 20× 0.1 Gy, compared to non-irradiated control tissue. (**B**) Quantitative whole-slide digital image analysis of nuclear H2A.J immunohistochemistry in lung parenchyma after fractionated IR (5 × 2 Gy; 24 h post-IR). Significant statistical difference compared to non-irradiated controls (marked by asterisk with square brackets): * *p* < 0.05.

## 4. Discussion

Chromatin organization and transcriptional profiles undergo tremendous rearrangements during senescence progression. To explore the potential epigenetic mechanisms of histone variant H2A.J in the regulatory networks of radiation-induced senescence, lung fibroblasts with down- and upregulated H2A.J expression levels were analyzed for their dynamic changes in chromatin organization, epigenetic and transcriptional landscape, and their cellular function following IR exposure. Through integrative ATAC-seq and RNA-seq analysis and functional validation of the senescence state, we identified and analyzed mechanistic interrelations between H2A.J incorporation and altered chromatin architecture with modified genome accessibility and transcription factor recruitment, changed SASP expression, and senescence competence. Our findings demonstrate that the fine-tuned incorporation of histone variant H2A.J defines the epigenetic landscape for driving the senescence programme in response to DNA-damaging stress. H2A.J overexpression, in contrast, was associated with reduced senescence-associated chromatin alterations, significant changes in the SASP secretome, and the potential to overcome senescence-associated growth arrest. Analyzing lung tissues following IR exposure, we observed increased H2A.J expression in specific cell populations (particularly in pneumocytes), most pronounced after high doses, but even after moderate and low doses. These findings suggest that H2A.J-associated epigenetic mechanisms play decisive roles even at therapeutic doses used during fractionated radiotherapy. It is increasingly recognized that DNA-damage triggered events stimulate inflammation and modulate cellular functions in injured normal tissues over months to years [31]. Future studies have to elucidate the precise role of H2A.J in triggering cytokine production during the temporal progression of lung injury, which manifests as acute pneumonitis and later as pulmonal fibrosis.

Collectively, our discoveries about the H2A.J-associated epigenetic mechanisms that drive radiation-induced damage development complement the traditional radiobiological conception of normal tissue injury. Understanding the precise molecular events responsible for the perpetuation of normal tissue injury is critical for developing therapeutic strategies to prevent, mitigate, and treat radiation toxicities.

Histone variants greatly expand the roles and dynamics of nucleosomes by wrapping more or less DNA, by having greater or lesser stability, through unique post-translational modifications, or by interacting with specific chromatin components [32]. Previous studies have shown that the incorporation of H2A.J, with its distinct sequence and properties compared to canonical H2A histone have direct effects on the structure and stability of nucleosomes [33]. Unlike canonical histones, which are distributed equally along the chromatin during replication, nucleosomes containing histone variants are deposited throughout the cell cycle at specific locations. Previous electron microscopic studies with H2A.J immunogold-labeling have shown that during radiation-induced senescence, H2A.J replaces canonical H2A preferentially at persisting DNA damage sites, and is enriched at the border of heterochromatic domains of DNA-SCARS arranged at the periphery of SAHF [14]. Our present findings suggest that fine-tuned H2A.J incorporation at chromatin sites involved in DDR processes is crucially important for establishing specific chromatin configurations and maintaining heterochromatin-euchromatin boundaries. Accordingly, deregulated H2A.J histone deposition in irradiated fibroblasts modifies local and global chromatin organization, and these structural changes impair the formation of SAHF and DNA-SCARS, thereby altering the epigenetic state of senescence. Increased H2A.J incorporation into senescent chromatin was also associated with the global loss of canonical histones, particularly for the different forms of the linker histone H1, which is known to play crucial roles in nucleosome positioning and heterochromatin formation [34].

The topological organization of nucleosomes across the genome regulate chromatin accessibility through a variety of chromatin mechanisms, such as altering TF binding by modulating steric hindrance. ATAC-seq measurements of open and closed chromatin accessibility reflect regulatory capacities and distinct functional states of chromatin organization [35]. Our findings indicate that the remodeling of senescence-related chromatin structures (SAHF and DNA-SCARS) modulates the physical access of transcription factors to chromatinized DNA. Our ATAC-seq analysis showed that AP-1 transcription factors, including members of the ATF, FOS, and JUN families, participate in the regulatory epigenome network of radiation-induced senescence. These findings confirm the previously described hierarchical structure of TF networks with AP-1 as the master regulator to choreograph transcriptional programmes via dynamic interactions with settler and migrant TFs during senescence progression [26]. In lung fibroblasts with properly regulated H2A.J expression, accessible chromatin sites for these senescence TFs were modified to the strongest degree following IR exposure, presumably to effectively regulate transcriptional activities for the timely execution of the senescence programme. H2A.J depletion in H2A.J-KD fibroblasts, by contrast, revealed notably reduced modulation capabilities for these chromatin accessibility sites following IR exposure, potentially implying their less flexible senescence regulation. H2A.J overexpression in H2A.J-KI fibroblasts revealed a lower open but a more closed TF interactome, potentially reflecting the disruption of TF senescence networks, with important implications for their senescent cell fate.

A substantial number of anticancer interventions, such as radiotherapy, induce senescence in normal but also in cancer cells by triggering genotoxic stress, leading to stable cell cycle arrest and SASP induction. Therefore, therapy-induced senescence serves as an initial antitumour mechanism to halt proliferation and prevent further genomic instability. However, we here present the first experimental evidence that H2A.J is progressively incorporated into the chromatin following IR exposure, and that H2A.J overexpression can reverse this radiation-induced cell cycle arrest. Therefore, H2A.J overexpression may promote tumorigenesis or even improve resistance to cancer treatments. Within the scope of our current studies on irradiated skin tissue, we observed that mitochondria-associated genes are also strongly modulated by H2A.J. Increasing evidence suggests that an altered mitochondrial metabolism is associated with the onset of resistance to radiotherapy [36]. However, the precise role of H2A.J in the context of mitochondrial metabolism and tumor response to radiotherapy has to be investigated in appropriate experimental studies.

Genome-wide chromatin accessibility profiling for TF-binding motifs using ATAC-seq emphasize the functional role of the histone variant H2A.J for regulatory senescence networks. Collectively, integrative analysis of ATAC-seq and RNA-seq datasets indicate that H2A.J-associated changes in chromatin accessibility in regulatory regions were clearly associated with modified inflammatory gene expression profiles. Our in vitro model with knock-down or knock-in of the H2A.J gene in a defined cell type makes it possible to study very precisely the functional effects of H2A.J on chromatin organization, the recruitment of transcription factors, and the pro-inflammatory secretome after IR exposure. However, fibroblasts are a rather radioresistant cell population that does not reflect the radiation reactions of diverse cell populations found in complex tissues. In future research work, knock-out mice will be examined before and after IR exposure in order to investigate the importance of H2A.J for the radiation response in the context of complex tissue homeostasis.

## 5. Conclusions

Cellular senescence is a stable cell growth arrest that is characterized by the silencing of proliferation-promoting genes through the compaction of chromosomes into SAHF [37,38]. Our findings suggest the differential incorporation of non-canonical histone variant H2A.J alter global chromatin architecture and as a consequence the epigenetic landscape. Different H2A.J deposition levels differentially modify the chromatin accessibility to transcription factors and selectively regulate the expression of inflammatory genes during radiation-induced senescence. Previous studies have shown that the depletion of H2A.J downregulates and the overexpression of H2A.J upregulates the expression of SASP components, but neither condition had any obvious effect on senescence-associated cell cycle arrest [13,14]. Here, we observed that the overexpression of H2A.J, in contrast, altered the senescence-associated proliferation arrest after IR exposure. These inconsistencies regarding the phenotypic effects of H2A.J overexpression can be explained by different analytical methods and by the different intensity and nature of senescence-inducing stressors. Our current findings suggest that the overexpression of H2A.J may impede heterochromatin formation following IR exposure and therefore inhibit the SAHF-mediated gene silencing of proliferation-promoting genes. DNA damage-induced senescence act as a potent anti-tumor mechanism by preventing the proliferation of potentially cancerous cells. Bypassing senescence and acquiring limitless replicative potential is a key event required for malignant transformation and for the development of radioresistence [39].

## Figures and Tables

**Figure 1 cells-12-00916-f001:**
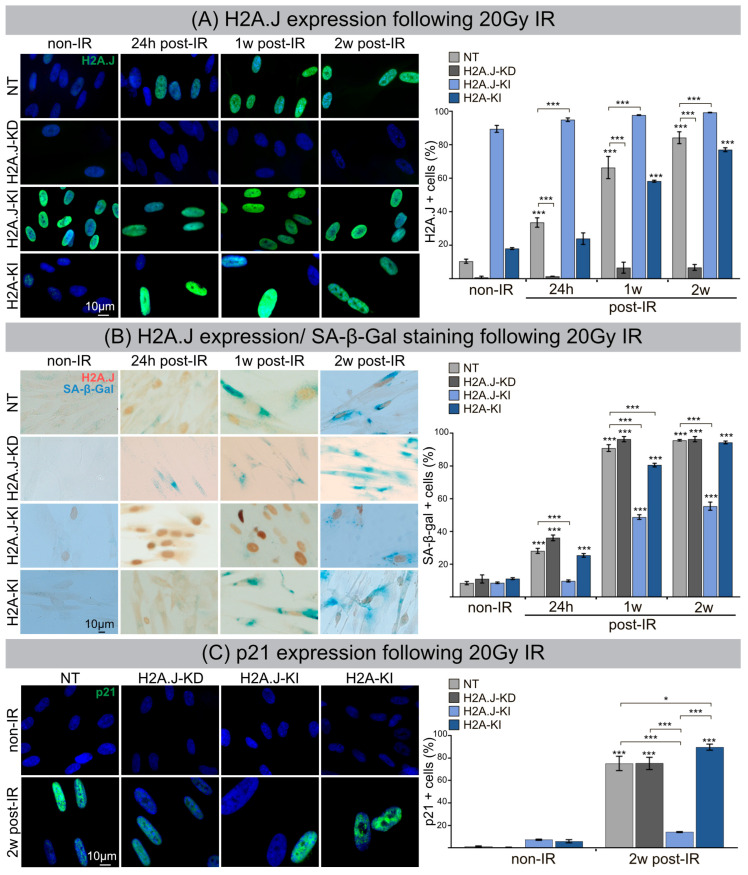
(**A**) H2A.J expression following 20 Gy IR. IFM micrographs of NT, H2A.J-KD, H2A.J-KI and H2A-KI fibroblasts show H2A.J+ cells at 24 h, 1 w and 2 w post-IR compared to non-irradiated fibroblasts. The adjacent graph shows the quantification of H2A.J+ cells. Data are presented as the mean of three experiments ±SE. Significant statistical difference compared to non-irradiated controls (marked by asterisks alone) or between cell lines (asterisks with square brackets): * *p* < 0.05; *** *p* < 0.001. (**B**) H2A.J expression and SA-β-Gal staining following 20 Gy IR. IHC micrographs show H2A.J and SA-β-Gal staining of NT, H2A.J-KD, H2A.J-KI and H2A-KI fibroblasts following 20 Gy at 24 h, 1 w and 2 w post-IR) compared to non-irradiated controls (non-IR). The adjacent graph shows the quantification of H2A.J+ and SA-β-Gal+ cells at 24 h, 1 w and 2 w post-IR, compared to non-irradiated controls. (**C**) p21 expression following 20 Gy IR. An IFM micrograph shows the p21 staining of NT, H2A.J-KD, H2A.J-KI and H2A-KI fibroblasts at 2 w post-IR compared to non-irradiated controls. The quantification of p21+ cells at 2 w post-IR compared to non-irradiated controls. Data are presented as the mean of three experiments ±SE. Significant statistical difference compared to non-irradiated controls (marked by asterisks alone) or between cell lines (asterisks with square brackets): * *p* < 0.05; *** *p* < 0.001.

**Figure 2 cells-12-00916-f002:**
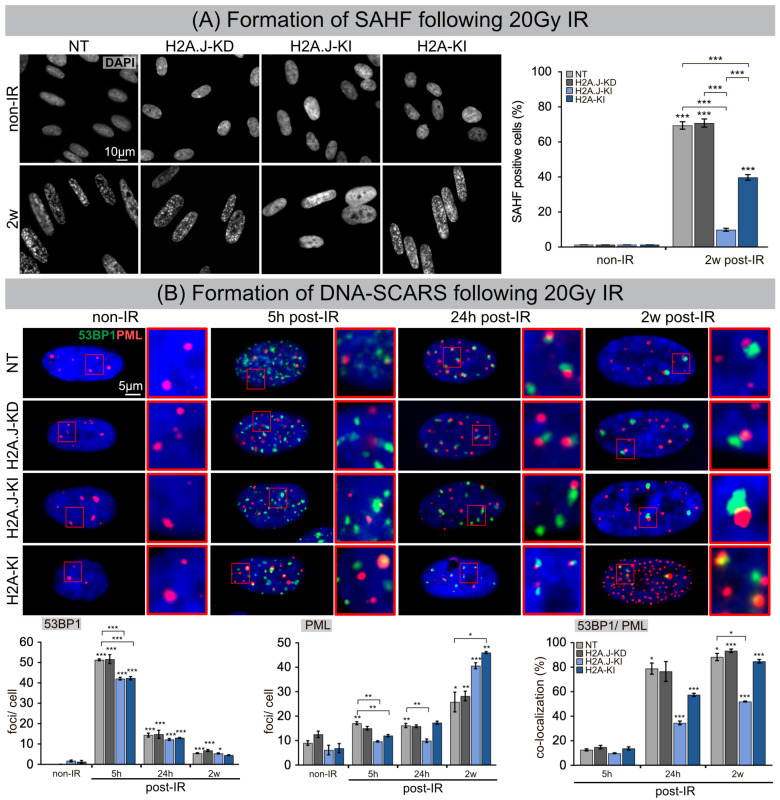
(**A**) Formation of SAHF following 20 Gy IR. IFM micrographs show the DAPI staining of NT, H2A.J-KD, H2A.J-KI and H2A-KI fibroblasts at 2 w post-IR, compared to non-IR controls. The adjacent graph shows the quantification of SAHF+ cells at 2 w post-IR compared to non-irradiated controls. Data are presented as the mean of three experiments ± SE. Significant statistical difference compared to non-irradiated controls (marked by asterisks alone) or between cell lines (asterisks with square brackets): *** *p* < 0.001. (**B**) Formation of DNA-SCARS following 20 Gy IR. IFM micrographs show double-staining for 53BP1 and PML in NT, H2A.J-KD, H2A.J-KI and H2A-KI fibroblasts at 5 h, 24 h and 2 w post-IR, compared to non-irradiated controls. The adjacent graph shows the quantification of 53BP1 foci/cell, PML-foci/cell and 53BP1/PML co-localization in NT, H2A.J-KD, H2A.J-KI and H2A-KI fibroblasts at 24 h, 1 w and 2 w post-IR. Significant statistical difference compared to non-irradiated controls (marked by asterisks alone) or between cell lines (asterisks with square brackets): * *p* < 0.05; ** *p* < 0.01; *** *p* < 0.001.

**Figure 3 cells-12-00916-f003:**
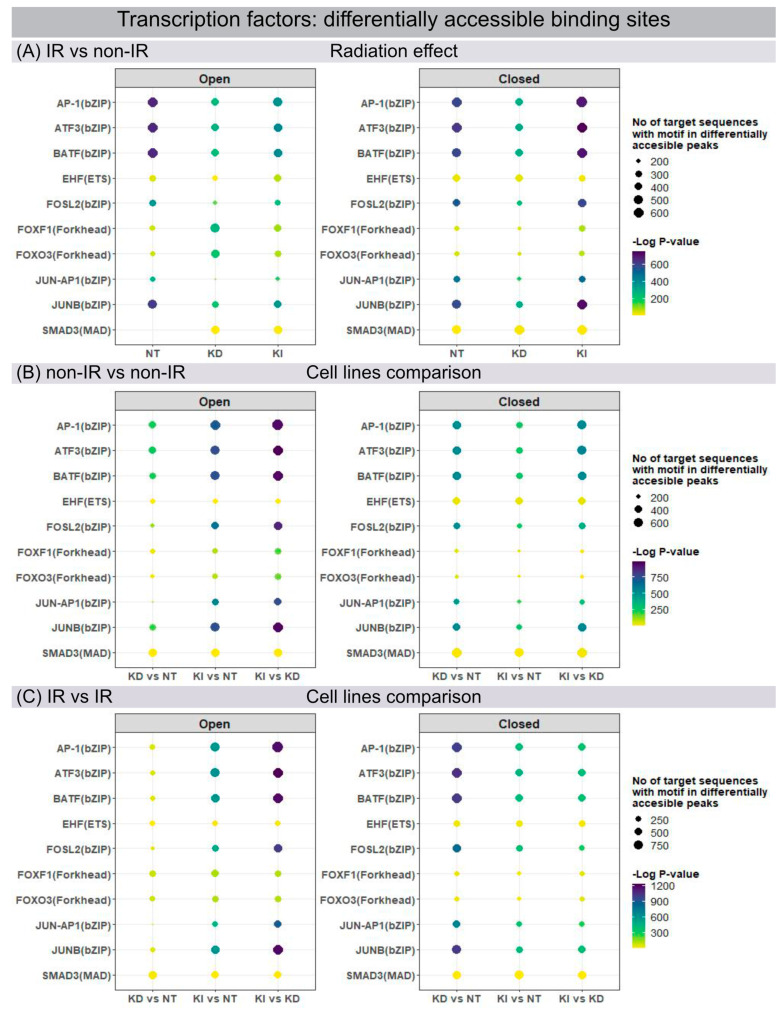
Genome-wide mapping of transcription factor binding sites. Changes in chromatin accessibility for transcription factor binding motifs in NT, H2A.J-KD and H2A.J-KI fibroblasts at 2 w post-IR, compared to their non-irradiated controls (**A**), and between NT, H2A.J-KD and H2A.J-KI fibroblasts in their non-irradiated (**B**) and their irradiated state (**C**).

## Data Availability

Research data are stored in an institutional repository and will be shared upon request to the corresponding author.

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
