# Peer review of "Role of Histone Variant H2A.J in Fine-Tuning Chromatin Organization for the Establishment of Ionizing Radiation-Induced Senescence"

_cells, 2023, doi:10.3390/cells12060916_

Round 1
Reviewer 1 Report
The article “Role of histone variant H2A.J in fine-tuning chromatin organization for the establishment of ionizing radiation-induced senescence” by Mutaz A. Abd Al-razaq, Benjamin M. Freyter , Anna Isermann, Gargi Tewary, Adèle Mangelinck, Carl Mann and Claudia E. Rübe describes rearrangements of chromatin organization and transcriptions during senescence; thereby the regulartory effects of the histone variant H2A.J are studied. Using lung fibroblasts and modifying the expression level of H2A.J the effects on chromatin dynamics, epigenetic changes and transcription are studied after ionizing radiation exposure. Mechanisms of altered genome accessibility are investigated. The results reveal that incorporation of H2A.J controls the epigenetic program of senescence in response to radiation induced DNA damage.
The article is well written and many details are well explained and shown in clear graphical representations and illustrative images. I recommend “accept after minor revision”.
1.) Typo: line 83: “…have shown…”
2.) In contrast to the biomolecular methods which are described in great detail, the description of statistical analysis is poor, reductive on the program applied. Please, extend this black box and explain the statistics used and why this is used for the data applied. What do statistical significances mean? Is p<0.05 etc. good or not enough. What is the null-hypothesis to which the p-values are calculated? Please, explain the statistics.
Author Response
1.) The error in line 83 has been corrected accordingly.
2.) GraphPad Prism (version 9.4.1, GraphPad Software, San Diego, CA, USA) was used to collect and analyse data. Data are presented as mean of three experiments ±SEM. One-way analysis of variance (ANOVA) with Dunnett’s multiple comparisons test was used for comparison among different groups. A p value of <0.05 was considered statistically significant, <0.01 as statistically highly significant and <0.001 as extremely statistically significant. Significant statistical differences compared to non-irradiated controls (marked by asterisks alone) or between cell lines (asterisks with square brackets) are presented in the figures as * (p <0.05), ** (p <0.01), and *** (p <0.001).
Reviewer 2 Report
Summary: An in vitro study evaluating the epigenetic mechanisms of H2A.J modifying transcription programmes during radiation-induced senescence progression.
Comments:
1. Methods section: the applied statistical tests required to be addressed. It is stated that the statistical analyses were done by GraphPad Prism. I suggested the authors to analyze the data using more dedicated software, such as SPSS, R, etc.
2. Fig 1A-right column. It is unclear whether the *** sign above H2A-KI bars is relative to which group—the similar case for Fig 1B and Fig 2.
Fig. 1B. Regarding 2w bars, In two cases ( H2A-KI vs. NT and H2A.J-KD vs. NT), there is a bit of difference between bar heights. Is the *** sign correct for all these cases?
3. As noted in the Conclusions section, senescence is a defensive mechanism for cancer cells to improve their resistance to cancer treatments. It is suggested the authors explain more about this issue in the Discussion section as the potential implication of their findings. The following information can help to improve this part of the Discussion section:
Recent evidence has demonstrated the crucial role of cancer cells' mitochondria in radiotherapy (https://pubmed.ncbi.nlm.nih.gov/36719474/) and immunotherapy resistance (https://pubmed.ncbi.nlm.nih.gov/36469835/). Yan et al. showed that H2A.J is one of the mitochondrial-related genes upregulated in breast cancer patients (https://pubmed.ncbi.nlm.nih.gov/33439992/). This finding addresses the potential role of H2A.J in breast cancer pathogenesis, possibly by activating mitochondria biogenesis.
4. Discussion section: Please add the study limitations, strengths, potential implications, and suggestions for future works.
Author Response
1.) GraphPad Prism (version 9.4.1, GraphPad Software, San Diego, CA, USA) was used to collect and analyse data. Data are presented as mean of three experiments ±SEM. One-way analysis of variance (ANOVA) with Dunnett’s multiple comparisons test was used for comparison among different groups. A p value of <0.05 was considered statistically significant, <0.01 as statistically highly significant and <0.001 as extremely statistically significant. Significant statistical differences compared to non-irradiated controls (marked by asterisks alone) or between cell lines (asterisks with square brackets) are presented in the figures as * (p <0.05), ** (p <0.01), and *** (p <0.001).
2.) Fig 1A-right column. It is unclear whether the *** sign above H2A-KI bars is relative to which group—the similar case for Fig 1B and Fig 2.
Basically, the asterisks directly above the bars (without additional brackets) show the statistically significant difference from the non-irradiated control of the same cell line (non-IR). These means for example, that for the cell line H2A-KI (dark blue bars): there is no statistically significant difference between 24h post-IR (no asteriks) and non-IR, but the values at 1w and 2w post-IR (each marked by 3 asterisk) are statistically highly different to the non-irradiated H2A-KI control (non-IR).
For the comparison between different cell lines, we use the brackets with asterisks to indicate which cell lines are being compared to each other. For example in Fig.1C the cell line H2A.J-KI (light blue bars): we observed for p21+ cells no statistical difference between 2w post-IR and non-IR (therefore no asteriks); but for the other cell lines (NT, H2A.J-KD and H2A-KI) we observed statistical differences to their non-irradiated controls (hence the asterisks directly above the bars). In addition, statistical significant differences were observed between the NT, H2A.J-KD and H2A-KI cell lines versus the H2A.J-KI cell lines, which are represented by the corresponding brackets with asteriks.
Fig. 1B. Regarding 2w bars, In two cases ( H2A-KI vs. NT and H2A.J-KD vs. NT), there is a bit of difference between bar heights. Is the *** sign correct for all these cases?
For each cell line the asterisks directly above the bars always indicate a statistical difference from the non-irradiated control (non-IR). Regarding 2w bars, there is no statistical difference between NT, H2A.J-KD and H2A-KI; but there is a statistical difference between NT and H2A.J-KI, presented by the bracket with asteriks.
3.) Substantial number of anticancer interventions, such as radiotherapy, induce senescence in normal but also in cancer cells by triggering genotoxic stress, leading to stable cell cycle arrest and SASP induction. Therefore, therapy-induced senescence serves as an initial antitumour mechanism to halt proliferation and prevent further genomic instability. However, here we present first experimental evidence that H2A.J is progressively incorporated into the chromatin following IR exposure and that H2A.J overexpression can reverse this radiation-induced cell cycle arrest. Therefore, H2A.J overexpression may promote tumorigenesis or even improve resistance to cancer treatments. Within the scope of our current studies on irradiated skin tissue, we observed that mitochondria-associated genes are also strongly modulated by H2A.J. Growing evidence suggests that altered mitochondrial metabolismus is associated with the onset of resistance to radiotherapy . However, the precise role of H2A.J in the context mitochondrial metabolism and tumor response to radiotherapy has to be investigated in appropriate experimental studies.
4.) Our in-vitro model with knock-down or knock-in of the H2A.J gene in a defined cell type makes it possible to study very precisely the functional effects of H2A.J on chromatin organization, recruitment of transcription factors and pro-inflammatory secretome after IR exposure. However, fibroblasts are a rather radioresistant cell population, that does not reflect the radiation reactions of diverse cell populations found in complex tissues. In future research work, knock-out mice will be examined before and after IR exposure in order to investigate the importance of H2A.J for the radiation response in the context of complex tissue homeostasis.
Round 2
Reviewer 1 Report
The authors followed all my suggestions for improvements. The manuscript can be accepted as it is.
Reviewer 2 Report
Dear authors
Thanks for addressing my comments. I have no more comments.